# Individual and Socioeconomic Contextual Factors Associated with Obesity in Brazilian Adolescents: VigiNUTRI Brasil

**DOI:** 10.3390/ijerph20010430

**Published:** 2022-12-27

**Authors:** Rafaella Lemos Alves, Natacha Toral, Vivian Siqueira Santos Gonçalves

**Affiliations:** 1Graduate Program in Human Nutrition, University of Brasília, Brasília 70910-900, DF, Brazil; 2Graduate Program in Public Health, University of Brasília, Brasília 70910-900, DF, Brazil

**Keywords:** food and nutrition surveillance system, adolescence, food consumption, eating behavior, primary health care

## Abstract

This study investigated the association of socioeconomic contextual factors of the municipality of residence of adolescents, their eating behavior and food consumption with the prevalence of obesity. This was a cross-sectional study, based on individual data regarding anthropometry, eating behavior (eating in front of screens and having at least three main meals a day), and markers of healthy and unhealthy eating of 23,509 adolescents aged 10 to 19 years, participating in the Food and Nutrition Surveillance of the Brazilian Population monitored in Primary Health Care (VigiNUTRI Brasil) assessment in 2018. Based on multilevel Poisson regression, a higher prevalence of obesity was observed among adolescents living in municipalities with per capita income above USD209.68 (PR = 1.22; 95% CI 1.05;1.42) and among those who consumed hamburgers and/or processed meats the previous day (PR = 1.09; 95% CI 1.01;1.17). Adolescents who had the habit of having three main meals a day (PR = 0.81; 95% CI 0.73;0.89 *p* < 0.05) and who consumed fresh fruit the previous day (PR = 0.91; 95% CI 0.84;0.98 *p* < 0.001) had lower prevalence of obesity. The results reinforce the influence of the social context and food consumption on obesity rates. The persistence of this situation implies a worsening in the current and future health of adolescents.

## 1. Introduction

Obesity is a critical public health problem that affects all stages of life, yet its prevalence has increased substantially among adolescents. According to the World Health Organization (WHO), between 1975 and 2016, the prevalence of obesity increased from 11 million in 1975 to 124 million in 2016 in children and adolescents aged 5 to 19 years [1]. Worldwide, more individuals are obese than underweight, except for sub-Saharan Africa and Asia [2]. In the United States, between 2017 to 2020, 14.7 million children and adolescents, aged 2 to 19 years old, were diagnosed with obesity [3]. In 2019, in Brazil, more than 620,000 adolescents aged 15 to 17 years had obesity [4].

The complexity of the disease treatment raises the probability of persistence of obesity in adulthood and the association with the early onset of other chronic noncommunicable diseases [5,6]. This fact is already a reality among Brazilian adolescents, as evidenced by the Study of Cardiovascular Risks in Adolescents (ERICA), revealing the existence of comorbidities such as hypertension, dyslipidemia, insulin resistance, and metabolic syndrome in this population [7,8,9,10].

Although poor diet remains one of the main risk factors responsible for this epidemiological context [11], social, demographic, cultural, and economic factors particularly influence the nutritional status of adolescents [12,13]. Thus, it is possible to make associations between contextual factors and indicators of the nutritional status of adolescents, in order to assess the contribution of risk factors for the development of obesity and implement appropriate corrective and preventive actions [14,15,16].

The use of public data made available by Health Information Systems (HIS) becomes a viable, easy-to-access, and low-cost alternative for providing continuous information on the nutritional status of adolescents. Among Brazilian computerized systems, the Food and Nutritional Surveillance System (Sisvan, in Portuguese) aims to provide data over time on the nutritional profile of the population using Primary Health Care (PHC) of the Unified Health System (SUS). Sisvan data assist in the formulation of actions, programs, and policies aimed at both the promotion of adequate and healthy eating, as well as the prevention and treatment of nutritional diseases [17].

However, analysis of information on the food and nutritional status of adolescents, obtained through Sisvan, is still little explored. Thus, given the relevance of using these data in view of the current epidemiological context, as well as the absence of national data on adolescents monitored in PHC, the aim of this study was to investigate the association of socioeconomic contextual factors of the municipality of residence of adolescents, eating behavior, and food consumption with the prevalence of obesity.

## 2. Materials and Methods

The present study was conducted in adherence with the Strengthening the Reporting of Observational Studies in Epidemiology (STROBE) statement recommendations [18].

### 2.1. Study Design

This is a cross-sectional study, from individualized data from the Sisvan regarding anthropometry, eating behavioral assessment, and food consumption of adolescents aged 10 to 19 years in the year 2018. This study integrates broader research, the Food and Nutrition Surveillance of the Brazilian Population followed in Primary Health Care (VigiNUTRI Brasil, in Portuguese). The Sisvan database was obtained through a specific request to the Ministry of Health, as recommended and provided in the Access to Information Law (Law No. 12,527 of 18/11/2011) and SAS/MS Ordinance No. 884 of 13/12/2011. 

### 2.2. Setting and Participants 

The VigiNUTRI Brasil analyzes data from follow-ups of individuals from all stages of life performed in Primary Health Care since 2008. For the present study, it selected adolescents with complete data of eating behavior, food consumption and anthropometry in the year 2018. 

### 2.3. Variables and Categories

The dataset records of adolescents for eating behavior and food consumption were obtained from the Sisvan Food Consumption Markers, that is an instrument completed by the Primary Health Care professional, through an interview with the adolescent [19]. First, it is asked if the adolescent has the habit of eating meals while watching television, a computer or cell phone (yes/no) and which are the meals usually eaten by him/her (options are breakfast, morning snack, lunch, afternoon snack, dinner and evening snack).

From the same instrument, information was obtained about food consumption, with questions regarding the previous day of the following markers of healthy eating (yes/no): beans; fresh fruit (not considering fruit juice); vegetables (not considering potato/cassava/yam); and markers of unhealthy eating (yes/no): hamburgers and/or processed meat (ham/salami/sausage); sweetened beverages (soda/juice box/powdered juice/coconut water box/syrups/fruit juice with added sugar); instant noodles, packaged snacks or salty cookies; and sandwich cookies or sweets (candies, lollipops, bubble gum, caramel, jelly).

The dataset records of adolescents also included anthropometric data of weight and height, that were measured by the Primary Health Care professional during the follow-up. For those who did not have both records on the same day (anthropometry and eating behavior/consumption), a difference of up to 30 days between them was accepted. Pregnant adolescents were not included in the database at the time of the evaluation.

#### 2.3.1. Obesity (Outcome)

From the anthropometric evaluation, the Body Mass Index-for-Age (BMI-for-Age) was obtained, which is calculated by dividing body mass in kilograms (Kg) and the square of height in meters (m²). The measurement of weight and height followed the "Guidelines for the collection and analysis of anthropometric data in health services: technical standard of the Food and Nutritional Surveillance System-SISVAN" [20]. The Z-score classification was based on the cutoff points established by the World Health Organization in 2007 and which are adopted by the Ministry of Health [20,21]. The following classifications were adopted: very low weight when Z-score < −3; underweight when Z-score < −2; eutrophy for the range of Z-score ≥ −2 and ≤+1; overweight for Z-score range > +1 and ≤ +2; obesity when Z-score > +2 [20,21]. Classification of obesity was considered as the outcome in the study analysis.

#### 2.3.2. Contextual Characteristics (Exposure Variables Related to Municipalities)

Four variables were considered as contextual characteristics, that were socioeconomic indicators from the adolescents’ municipalities of residence: the degree of urbanization; the percentage of population covered by the Family Health Strategy; the average per capita income; and the Gini index. 

The Urbanization Degree of the municipalities, year 2017, was obtained from the Brazilian Institute of Geography and Statistics (IBGE) [22] and classified as: Low (population units with less than 50% of the population residing in densely occupied areas), Moderate (population units with 50 to 75% of the population residing in densely occupied areas) and High (population units with more than 75% of the population residing in densely occupied areas) [23].

The percentage of population covered by the ESF (% ESF) teams for the year 2018 was obtained from a public consultation, through the Secretariat of Primary Care of the Ministry of Health in the space of information and access to Primary Care systems-e-Gestor Basic Care [24] and classified by percentiles: < 74.36 (percentile < 25); ≥74.36 and <100 (percentile ≥ 25 and percentile ≤ 75); = 100 (percentile > 75). The ESF is the primary health care organization model in Brazil, which reorients the work process in accordance with SUS principles and guidelines [25]. 

The average per capita income (in American dollars) of the municipalities, year 2017, was classified according to the cutoff points obtained by percentiles as: <92.03 (percentile < 25); ≥92.03 and ≤209.68 (percentile ≥ 25 and percentile ≤ 75); >209.68 (percentile > 75) [22]. The Gini Index, year 2010, was classified according to the following percentiles: <0.47 (percentile < 25); ≥0.47 and ≤0.54 (percentile ≥ 25 and percentile ≤ 75); >0.54 (percentile > 75). The closer the Index is to 1, the greater the income inequality in the municipality. Data of the per capita income and Gini Index were obtained from the Atlas of Human Development in Brazil [26], which is an electronic database based on microdata from the year 1991.

#### 2.3.3. Individual Characteristics-Exposure Variables Obtained from Sisvan

Having or not having the habit of eating meals in front of screens and the habit of eating at least three main meals a day (breakfast, lunch and dinner) were considered as eating behavior variables. The markers of food consumption for healthy (bean consumption; consumption of fresh fruit; and consumption of vegetables) and unhealthy eating (consumption of hamburgers and/or processed meat; consumption of sweetened beverages; consumption of instant noodles, packaged snacks or salty cookies; and consumption of sandwich cookies or sweets) were also listed as exposure variables [19]. 

#### 2.3.4. Individual Characteristics-Covariates

The demographic characteristics analyzed, considered as confounders in the study analysis, were sex, age (10 to 19 years), which was obtained from the difference between the date of care and the date of birth of the adolescent, and participation in the Bolsa Família Program (PBF) (yes/no). 

The PBF is a cash transfer program of the Brazilian federal government for families with low socioeconomic status [27]. In 2021, the PBF was renamed Programa Auxílio Brasil.

### 2.4. Data Cleaning

Some care was taken to avoid measurement errors. As it is a secondary database, it was necessary to make adjustments in the displacement of information, verification of missing data, exclusion of duplicate observations, exclusion of participants outside the age range of the study, standardization of weight and height variables, analysis of outliers, calculation of the Z-score for height, weight and BMI-for-age. The database consistency analysis was performed by a trained professional with extensive experience in the area.

### 2.5. Statistical Methods

The outcome of interest was obesity, and the exposure variables were those of eating behavior and food consumption, in addition to the variables of the municipalities. To investigate this association, we started from a theoretical model in which municipal indicators and markers of eating behavior and food consumption could be associated with obesity. The variables related to participation in the PBF (proxy of socioeconomic status), sex and age were used for model adjustment (Figure 1).

Descriptive statistics were used to calculate the relative and mean frequency of the variables, and the prevalence of obesity was estimated with the respective 95% confidence interval (CI). Multilevel mixed-effects Poisson regression was the method used to assess the association between the variables of interest. This method considers both fixed effects and random effects, which is important because random effects are useful for modeling intracluster correlation. That is, the models considered that observations in the same cluster (municipality) were correlated, because they shared common cluster-level random effects [28]. The analysis started with the construction of an empty model, without the inclusion of independent variables, to obtain the variance related to the outcome attributed to the municipality. Then, the independent variables were tested individually in relation to the outcome and those with *p* < 0.20 were selected for multivariate analysis. Four models were built, the first with only characteristics of the municipality (contextual level), the second with characteristics related to eating behaviors, the third related to food consumption markers (individual level) and the fourth aggregating the two levels. Variables that were not significant in models 1, 2 and 3 were not included in model 4. The multivariate analysis was adjusted for sex, age and participation in the PBF. The association between variables was calculated through the crude and adjusted prevalence ratio (PR) and 95%CI, with a significance level for the tests of 5% (*p* < 0.05). Data were analyzed using Stata software, version 17.0. Regression models were built using the "mepoisson" command (Statacorp, College Station, TX, USA). 

### 2.6. Ethical Aspects

This study was approved by the Ethics and Research Committee of the Faculty of Health Sciences, University of Brasilia, under CAAE number 19024819.3.0000.0030. 

## 3. Results

The records of 23,509 adolescents followed up in Primary Health Care in 2018 were evaluated. The comparison between cases obtained from Sisvan and population estimates carried out by IBGE for the reference year, 2018, by Brazilian macro-region, is presented in Table 1.

It was found that most adolescents lived in municipalities with a low degree of urbanization and with 100% ESF coverage. The mean per capita income and Gini Index were USD161.00 (±77.55) and 0.504 (±0.064), respectively. Most were female and aged between 10 and 13 years, 12.74% (95% CI 12.31;13.17) had obesity and 85.03% (95% CI 84.57;85.48) were not registered in the PBF (Table 2).

Regarding eating behavior, 66.81% (95% CI 66.21;67.41) of the adolescents had meals in front of the screens and 44.81% (95% CI 44.18;45.45) did not have their three main meals a day (breakfast, lunch and dinner). Regarding food consumption, around 15% did not eat beans and more than 30% did not have fruits or vegetables the day before. At least 43% of the adolescents ate one of the investigated markers of unhealthy eating the day before (Table 3). 

Higher prevalence of obesity was seen in adolescents who lived in municipalities with a high degree of urbanization, with ESF coverage lower than 74.36% and where municipal per capita income was higher than USD179.34 (14.99–95% CI 14.10;15.92) (Table 4). No association was observed with Gini index.

There was a high prevalence of adolescents with obesity (14.17%) among those who do not have the habit of eating three main meals a day (95% CI 13.58;14.78). Adolescents who did not consume beans (14.24–95% CI 13.12;15.43), fresh fruits (13.98–95% CI 13.20;14.80) and vegetables (13.56–95% CI 12.82;14.34) the previous day had a higher prevalence of obesity compared with those without the disease. Similarly, adolescents who consumed hamburgers and/or other processed meats the previous day (13.35–95% CI 12.705;14.02) had higher prevalence of obesity (Table 5). No association was observed with having meals in front of screens or the other investigated food consumption markers. 

From the regression analyses, statistically significant associations (*p* < 0.05) with obesity were observed among adolescents living in municipalities with per capita income above USD209.68 (PR = 1.22–95% CI 1.05;1.42) and among those adolescents who consumed hamburgers and/or processed meats the previous day (PR = 1.09–95% CI 1.01; 1.17). On the other hand, adolescents who had the habit of having the three main meals a day (PR = 0.81–95% CI 0.73;0.89 *p* < 0.05) and who consumed fresh fruit the previous day (PR = 0.91–95% CI 0.84;0.98 *p* < 0.001) had a lower prevalence of obesity (Table 6). 

## 4. Discussion

The present study was the first to analyze individual data from Sisvan of adolescents followed up in PHC in Brazil. The highest prevalence of obesity was associated with higher municipal per capita income and consumption of hamburgers and/or processed meats the day before. In contrast, lower prevalence of obesity was associated with having the habit of eating three main meals a day and eating fresh fruit the day before. Furthermore, the results indicate that despite the higher prevalence of consumption of healthy eating markers, there was a significant consumption of markers of unhealthy eating and a high prevalence of the habit of having meals in front of screens.

Inadequate food consumption and increased prevalence of obesity among Brazilian adolescents have been frequently observed in recent population surveys conducted in the country. The 2017−2018 Household Budget Survey (POF) pointed out that consumption of fruits and vegetables was lower among adolescents compared to older age groups. As an aggravating factor, the high consumption of markers of unhealthy eating (such as sandwich cookies, soft drinks, dairy drinks, packaged snacks, sandwiches, pizzas) by adolescents stands out [29]. In the 2019 National School Health Survey (PeNSE), 97.3% of adolescents consumed at least one ultraprocessed food (UPF) the day before the survey [30]. As in population surveys, there was high consumption of markers of unhealthy eating the day before by adolescents followed up in PHC in 2018. 

As for the presence of obesity, our results (12.74–95% CI 12.31;13.17) are close to the data from the National Health Survey, which found a prevalence of 13.4% of adolescents aged 15–17 years with obesity [4]. Since our data refer to PHC adolescents, who are generally more vulnerable, we can highlight the potential of using Sisvan to monitor the health of the population in this age group, as they are closer to the general population.

The prevalence of obesity demonstrates that we are facing a public health problem that has worsened over time, and that is a reflection of the food and nutrition transition process. Society has become increasingly urban and has experienced new demographic, behavioral, and consumption patterns [31,32]. The environment where the individual lives, and the different contextual factors that surround him/her, are recognized in the literature as risk factors for the development of diseases, especially chronic non-communicable diseases, such as obesity examined here. In a study of Chinese children and adolescents aged 6 to 17 years, higher rates of overweight and obesity were observed in urban areas and in families with higher income. In addition, the authors highlighted the higher prevalence in males [12].

Although this study considered aggregated rather than individual contextual data, limiting comparison with the previous study, the results corroborate what was observed by Fan and Zhang (2021), with data from 137,995 adolescents aged 12 to 15 years from 21 countries based on Global School-Based Student Health Surveys from 2005 to 2017. The authors observed a trend of significant increase in the prevalence of being overweight and obesity, particularly in high-, upper-middle-, and lower-middle-income countries [13]. 

There is evidence that low socioeconomic status is associated with higher BMI in high-income countries, and low BMI in middle-income countries [32]. It is noteworthy that 14.97% of adolescents assessed in this study were beneficiaries of the former PBF. 

In Brazil, there is great sociocultural, economic and urbanization inequality between geographic regions. The country has many developed cities and regions, in terms of urban infrastructure, basic sanitation, work, housing, access to health and education; on the other hand, it is possible to find others with much lower development rates, which impacts the lifestyle of young people, especially in relation to behavior and healthy food consumption [33,34,35].

This scenario demonstrates the importance of systematic and constant monitoring of the nutritional status and dietary profile of adolescents, since in this age group there is the consolidation of eating behavior, with a great diversity and complexity of food choices, which, associated with body changes inherent to growth and sexual maturation, can lead to excessive weight gain [6,36]. Excess weight gain in the transition from adolescence to adulthood increases the risk of obesity in young adults. According to Kartiosuo et al. (2019), obese adolescents are 89.0% more likely to become obese adults [37]. Similar data was found in the meta-analysis study carried out with 15 prospective cohort studies, where it was shown that about 80.0% of obese adolescents will still have obesity in adulthood and about 70.0% will have obesity after 30 years [38]. 

The eating behavior of adolescents has been highlighted in different studies. Data from ERICA showed that more than half of adolescents had meals almost always or always in front of the TV, and consumed snacks in front of screens with this same frequency [34]. Eating in front of screens, as well as spending more time in front of screens, is associated with higher caloric contribution of UPF in adolescents’ diet and sedentary lifestyle [36,39]. 

According to national and international studies, it is difficult for young people to adopt and maintain healthy eating patterns during adolescence. In the Longitudinal Study of Nutritional Assessment (ELANA), carried out with 1,035 adolescents from six schools in the metropolitan region of Rio de Janeiro, adolescents who consumed UPF tended to have a lower daily intake of fruits, raw and cooked vegetables and a higher intake of sugar [40]. Doggui et al., (2021) evaluated 744 Canadian adolescents (age 11 to 18 years), followed from 2013 to 2019, and observed a decline in daily breakfast consumption, low consumption of fruits and vegetables, and increased consumption of fast food [6].

A previous study with data from ERICA highlighted that the UPF most consumed by adolescents were instant pasta, packaged cookies, soft drinks and processed meat, and that higher consumption of these foods was associated with not having breakfast regularly [36]. In our final model, presented in this work, having the three main meals a day (breakfast, lunch and dinner) and the consumption of fruits the day before were associated with a lower prevalence of obesity. 

Our findings are in agreement with the recommendation of the Food Guide for the Brazilian Population, to have at least three main meals, since these represent about 90% of the total calories consumed throughout the day [41]. This result is believed to be unprecedented in the literature, since it differs from most analyses carried out by other studies with adolescents.

The contribution of having meals instead of snacks to reduce the prevalence of obesity has already been observed in studies that evaluated adherence to school meals by adolescents [42,43]. Adolescents with high adherence to school meals (5 times/week) had 11.00% less prevalence of overweight (PR = 0.89–95% CI 0.80;0.99) and 24.00% less prevalence of obesity (PR = 0.76−95% CI 0.62;0.93) than those with lower adherence (<5 times/week) [43].

In general, it can be considered that the fact that the traditional Brazilian food pattern (consumption of rice, beans, among other foods) is present in the daily life of the population contributes to the prevention of obesity, since it is characterized as a more nutritious diet compared to UPF [44,45]. In a study with seven cohorts of Swedish adolescents, high fruit intake was associated with higher vegetable intake (OR=25.7–95% CI 20.0;33.1), and a higher frequency of vegetable consumption was associated the lowest chance of being overweight and obesity among them (OR=0.77–95% CI 0.62–0.95) [46]. In Norway, researchers did not find any beneficial effect on the BMI of children and adolescents who consumed fruits and vegetables in schools for up to four years [47].

It is not possible to infer that the consumption of fruits and vegetables, by itself, has a protective effect against the disease; however, it should be encouraged and guaranteed by public policies for children and adolescents, especially those with lower family income, in order to guarantee the daily consumption of fiber, vitamins, minerals and antioxidant compounds. In this regard, Brazil is a protagonist in the formulation of public policies that aim to reduce the impact of the burden of chronic diseases on the population and improve the quality of life, through food and nutrition actions [5,48].

On the other hand, the adherence to eating patterns consisting of snacks and fast food by Brazilian adolescents showed a high correlation with the consumption of ham, bologna, turkey breast and salami, among other UPF groups [45]. In the same study, young people who adhered to these standards were more likely to be overweight (OR = 1.50–95% CI 1.13–1.99 for the fifth quintile of the snacks pattern; OR = 1.55–95% CI 1.12–2.12 for the fifth quintile of the fast-food standard) [45].

Kelly et al. (2022) in their study addressing the role of the gut-brain axis and its interaction with reward systems from UPF consumption, point out that understanding the different food choices is a critical problem in health research [49]. However, the main points of discussion about the choice and consumption of UPF are related to its formulation, which deconstructs whole foods into chemical constituents, altering and recombining them with additives in products composed of low diversity of nutrients and highly palatable [41]. Another property of UPFs is the supply of usable calories quickly becoming a potential factor driving their overconsumption [49]. The mechanisms of action of this consumption, which lead to adverse health implications, may be the result of nutrient content, or ultra-processing [49,50]. 

The fact is that the amount of scientific evidence proving the association of UPF consumption with chronic non-communicable diseases and negative health outcomes has grown. In a systematic review and meta-analysis, five prospective cohort studies showed that higher UPF consumption was associated with an increased risk of all-cause mortality (RR = 1.25–95% CI, 1.14, 1.37; *p* < 0.00001), three studies showed an increased risk of cardiovascular disease (RR = 1.29−95% CI, 1.12;1.48; *p* = 0.0003), two studies showed a risk for cerebrovascular disease (RR = 1.34−95% CI, 1.07;1.68; *p* = 0.01) and two studies for depression (RR = 1.20–95% CI, 1.03;1.40; *p* = 0.02) [51]. Another review with prospective cohort studies showed that higher UPF intake increases the risk of obesity, cardiovascular disease, cancer and type 2 diabetes [52]. Therefore, actions are needed to help reduce the consumption of UPF by the population and promote an adequate and healthy diet.

The consumption of hamburgers and processed meats was associated with the highest prevalence of obesity in our study. In the systematic review and meta-analysis conducted by Rouhani et al., (2014), it was shown that red and processed meat intake was directly associated with risk of obesity, higher BMI and waist circumference. In addition, consumption of processed meat has been also proved as a cause of colorectal cancer [53]. 

The markers of eating behavior and food consumption of Sisvan are not routinely recorded by professionals, who prioritize recording anthropometric measurements. This questionnaire of food consumption markers was included in Primary Health Care routine in 2015, but many professionals are still reluctant to incorporate this practice as a habit, generating less data on the population’s diet. Furthermore, food consumption data are based only on the day before the consultation and the anthropometric assessment based only on BMI does not reflect body composition. However, despite these possible variations, these instructions are reinforced in the national guideline for data collection and measurements of Sisvan in Primary Health Care in Brazil [20].

Other limitations must be taken into account, such as the fact that a secondary database is used, where information may be underreported, errors may occur in the collection, typing and storage of data, reflecting the representativeness of the sample. Similar to all studies with secondary data from health services, our sample does not have representativeness and it is not an intentional study. Despite these limitations, a comparison was made between the analyzed sample and the projection of the Brazilian population according to the IBGE, in the year 2018, observing a similar distribution between the macroregions. For the adolescents’ data of Sisvan, the Southeast and Northeast regions were the ones that most contributed to the total sample, as well as the contingent estimated by the IBGE, whose largest projected populations for the same age group would be in these regions. Sisvan data cover all Brazilian regions, and the system is available to all Brazilian municipalities, which makes its capillarity of great relevance for public health actions in Brazil.

The cross-sectional design of the study prevents the inference of causality in the associations found. However, the way in which the data were obtained, and the consistency analysis performed, with the removal of duplicate follow-ups, missing data or inconsistent data on age, BMI calculations and nutritional status classification, were taken care of to minimize possible biases, thus, it may be considered a strength of our study. In addition, the cross-sectional design is a starting point for future investigations that will be able to carry out more robust analyses based on the results found.

## 5. Conclusions

The results reinforce the association of social context and food consumption with the rate of obesity among adolescents. The persistence of this scenario implies worsening in the current and future health of adolescents. Thus, it is essential to make better use of HIS data, such as Sisvan, as an efficient tool for periodic use to contribute to the strengthening of food and nutrition actions, better planning and management of public health systems, and the encouragement of programs and public policies directed to obesity management in this stage of life. 

## Figures and Tables

**Figure 1 ijerph-20-00430-f001:**
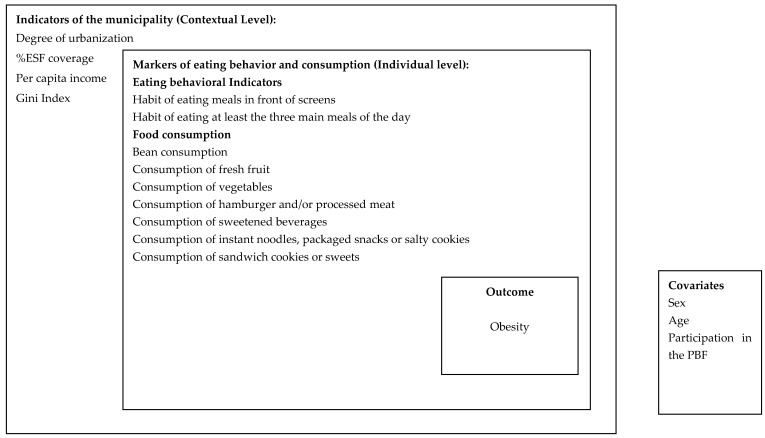
Theoretical Model with the variables proposed for cross-sectional study analysis.

**Table 1 ijerph-20-00430-t001:** Adolescents aged 10 to 19 years followed up on Food and Nutrition Surveillance System (Sisvan) and IBGE population estimate of adolescents and macro-region, Brazil, 2018.

Macro-Region	Sisvan Sample	Population Estimated by IBGE
	*n* (%)	*n* (%)
Midwest	1381 (5.87)	2,447,144 (7.74)
Southeast	11,020 (46.88)	12,085,577 (38.22)
Northeast	5449 (23.18)	9,564,775 (30.25)
North	2970 (12.63)	3,445,037 (10.89)
South	2689 (11.44)	4,079,337 (12.90)
Brazil	23,509	31,621,870

**Table 2 ijerph-20-00430-t002:** Characterization of adolescents followed by the Food and Nutrition Surveillance System (Sisvan), Brazil, 2018.

Variables	% or Average	95% CI or SD
**Contextual Variables**		
**Degree of urbanization**		
Low degree of urbanization	38.15	37.53–38.77
Moderate degree of urbanization	26.01	25.45–26.58
High degree of urbanization	35.84	35.23–36.45
**ESF Coverage**		
<74.36%	24.84	24.29–25.39
≥74.36% e < 100%	16.21	15.74–16.68
=100%	58.96	58.32–59.58
** Per capita income (dollar)**	161.00	77.55
**Gini Index**	0.504	0.064
**Individual Characteristics**		
** Sex**		
Male	35.11	34.50–35.72
Female	64.89	64.27–65.50
**Age**		
10 to 13 years old	43.53	42.90–44.17
14 to 16 years old	26.44	25.88–27.01
17 to 19 years old	30.00	39.44–30.61
**Nutritional Status**		
Very low weight	0.84	0.73–0.97
Low weight	3.59	3.36–3.83
Eutrophy	63.24	62.62–63.86
Overweight	19.59	19.08–20.10
Obesity	12.74	12.31–13.17
**PBF Participation**		
Yes	14.97	14.52–15.43
No	85.03	84.57–85.48

Legend: CI: confidence interval; SD: standard deviation; degree of urbanization of the municipalities for the year 2017; %ESF coverage-percentage of coverage of the Family Health Strategy for the year 2018; per capita income of the municipality for the year 2017; Gini index for the year 2010. Dollar exchange rate in December 2018: R$3.87 (Brazilian reais). Degree of urbanization: *n* = 23,471. %ESF coverage: *n* = 23,509. Per capita income; Gini index: *n* = 23,470. PBF participation-participation in the Bolsa Família Program. Individual characteristics: *n* = 23,509.

**Table 3 ijerph-20-00430-t003:** Indicators of eating behavior and food consumption of adolescents followed by the Food and Nutrition Surveillance System (Sisvan), Brazil, 2018.

	%	95% CI
Eating Behavior		
Have meals in front of screens		
Yes	66.81	66.21–67.41
No	31.03	30.45–31.63
Don’t know	2.15	1.97–2.35
**Have three main meals a day**		
Yes	55.19	54.55–55.82
No	44.81	44.18–45.45
**Previous Day’s Consumption**		
**Beans**		
Yes	83.97	83.49–84.43
No	14.94	14.49–15.40
Don’t know	1.09	0.97–1.23
**Fresh fruit**		
Yes	68.13	67.53–68.72
No	30.58	29.99–31.17
Don’t know	1.29	1.15–1.44
**Vegetables**		
Yes	65.56	64.95–66.16
No	33.14	32.54–33.75
Don’t know	1.30	1.16–1.45
**Hamburger and/or processed meat**		
Yes	43.29	42.66–43.93
No	55.09	54.45–55.72
Don’t know	1.62	1.46–1.79
**Sweetened Beverages**		
Yes	65.62	65.01–66.22
No	32.89	32.23–33.49
Don’t know	1.50	1.35–1.66
**Instant noodles/packaged snacks/salty cookies**		
Yes	44.46	43.83–45.1
No	53.89	53.26–54.53
Don’t know	1.64	1.49–1.81
**Sandwich cookies or sweets**		
Yes	54.59	53.95–55.23
No	43.67	43.03−44.30
Don’t know	1.74	1.58–1.91

*n* = 23,509. Legend: CI: confidence interval; Habit of eating meals in front of screens (while watching TV, playing computer and/or cell phone); Habit of eating at least the three main meals of the day (breakfast, lunch and dinner); Fresh fruit (not considering fruit juice); Vegetables (not considering potato/cassava/yam); Hamburgers and/or processed meat (ham/salami/sausage); Sweetened beverages (soda/juice box/powdered juice/coconut water box/syrups/fruit juice with added sugar); Sandwich cookies or sweets (candies, lollipops, bubble gum, caramel, jelly).

**Table 4 ijerph-20-00430-t004:** Prevalence of obesity according to characteristics of the municipalities of residence of adolescents followed by the Food and Nutrition Surveillance System (Sisvan), Brazil, 2018.

	Obesity	
	%	95% CI
**Degree of urbanization**		
Low degree of urbanization	12.16	11.50–12.85
Moderate degree of urbanization	11.55	10.77–12.37
High degree of urbanization	14.20	13.48–14.97
**ESF Coverage**		
<74.36%	14.68	13.79–15.61
≥74.36% e < 100%	12.33	11.33–13.42
= 100%	12.03	11.50–12.58
**Per capita income (dollar**)		
<92.03	11.42	10.63–12.26
≥92.03 e ≤ 209.68	12.25	11.67–12.86
>209.68	14.99	14.10–15.92
**Gini Index**		
<0.47	12.74	11.87–13.65
≥0.47 e ≤ 0.54	13.19	12.57–13.84
>0.54	12.01	11.28–12.79

Legend: CI: confidence interval; degree of urbanization of the municipalities referring to the year 2017; %ESF coverage - percentage of coverage of the family health strategy referring to the year 2018; per capita income referring to the year 2017; Gini index referring to the year 2010. Dollar exchange rate in December 2017: R$3.31. Degree of urbanization: *n* = 23,471. %ESF coverage: *n* = 23,509. Per capita income; Gini index: *n* = 23,470.

**Table 5 ijerph-20-00430-t005:** Prevalence of obesity according to indicators of eating behavior and food consumption of adolescents followed by the Food and Nutrition Surveillance System (Sisvan), Brazil, 2018.

	Obesity	
	%	95% CI
**Eating Behavior**		
**Have meals in front of screens**		
Yes	12.82	12.31–13.35
No	12.25	11.52−13.02
Don’t know	16.99	13.97–20.52
**Have three main meals a day**		
Yes	10.97	10.39–11.58
No	14.17	13.58–14.78
**Previous Day’s Consumption**		
**Beans**		
Yes	12.43	11.98–12.90
No	14.24	13.12–15.43
Don’t know	15.56	11.63–20.52
**Fresh fruit**		
Yes	12.14	11.64–12.65
No	13.98	13.20–14.80
Don’t know	14.85	11.27–19.31
**Vegetables**		
Yes	12.29	11.78–12.81
No	13.56	12.82–14.34
Don’t know	14.10	10.62–18.47
**Hamburger and/or processed meat**		
Yes	13.35	12.705–14.02
No	12.13	11.58–12.704
Don’t know	16.80	13.37–20.89
**Sweetened Beverages**		
Yes	12.55	12.04–13.08
No	12.97	12.24–13.74
Don’t know	15.62	12.19–19.80
**Instant noodles/packaged snacks/salty cookies**		
Yes	12.44	11.82–13.08
No	12.84	12.27–13.43
Don’t know	17.36	13.90–21.46
**Sandwich cookies or sweets**		
Yes	12.53	11.97–13.11
No	12.89	12.25–13.55
Don’t know	15.40	12.22–19.23

Legend: CI: confidence interval; Habit of eating meals in front of screens (while watching TV, playing computer and/or cell phone); Habit of eating at least the three main meals of the day (breakfast, lunch and dinner); Fresh fruit (not considering fruit juice); Vegetables (not considering potato/cassava/yam); Hamburgers and/or processed meat (ham/salami/sausage); Sweetened beverages (soda/juice box/powdered juice/coconut water box/syrups/fruit juice with added sugar); Sandwich cookies or sweets (candies, lollipops, bubble gum, caramel, jelly). Have meals in front of screens and Food Consumption Indicators: *n* = 23,509; Having three main meals a day: *n* = 11,928.

**Table 6 ijerph-20-00430-t006:** Contextual and individual factors associated with obesity in adolescents followed by the Food and Nutrition Surveillance System (Sisvan), Brazil, 2018.

Features	Crude Analysis		Model 1		Model 2		Model 3		Model 4	
Contextual Level	PR	95% CI	PR	95% CI	PR	95% CI	PR	95% CI	PR	95% CI
**Degree of Urbanization**										
Low degree of urbanization	Ref.									
Moderate degree of urbanization	0.95	0.83–1.09								
High degree of urbanization	1.10 *	0.97–1.24								
**%ESF coverage**										
<74.36	Ref.									
≥74.36 e < 100	0.871 *	0.73–1.03								
=100	0.874 *	0.77–1.00								
**Per capita income (dollar)**										
<92.03	Ref.		Ref.						Ref.	
≥92.03 e ≤ 209.68	1.10	0.95–1.27	1.10	0.95–1.27					1.08	0.94–1.23
>209.68	1.28 ***	1.10–1.49	1.26 **	1.05–1.51					1.22 **	1.05–1.42
**Gini Index**										
<0.47	Ref.									
≥0.47 e ≤ 0.54	1.09 *	0.95–1.25								
>0.54	1.00	0.86–1.16								
**Individual Level**										
**Have meals in front of screens**										
No	Ref.									
Yes	1.04	0.96–1.13								
**Have three main meals a day**										
No	Ref.				Ref.				Ref.	
Yes	0.83 ***	0.76–0.92			0.80 ***	0.73–0.88			0.81 ***	0.73–0.89
**Previous Day’s Consumption**										
**Beans**										
No	Ref.									
Yes	0.92 *	0.83–1.02								
**Fresh fruit**										
No	Ref.						Ref.		Ref.	
Yes	0.90 ***	0.83–0.97					0.91 **	0.84–0.99	0.91 **	0.84–0.98
**Vegetables**										
No	Ref.									
Yes	0.94 *	0.88–1.02								
**Hamburger and/or processed meat**										
No	Ref.						Ref.		Ref.	
Yes	1.09 **	1.01–1.18					1.09 **	1.01–1.18	1.09 **	1.01–1.17
**Sweetened Beverages**										
No	Ref.									
Yes	0.96	0.89–1.04								
**Instant noodles/packaged snacks/salty cookies**										
No	Ref.									
Yes	0.97	0.90–1.05								
**Sandwich cookies or sweets**										
No	Ref.									
Yes	0.95	0.89–1.03								

Legend: PR: prevalence ratio (PR); CI: confidence interval; Model 1, contextual variables; Model 2, individual variables of eating behavior; Model 3, individual variables of food consumption; Model 4, contextual and individual variables. All models were adjusted for sex, age, and participation in the Bolsa Família Program. * *p* < 0.20, ** *p* < 0.05, *** *p* < 0.001.

## Data Availability

All data supporting the results of this study can be obtained upon request to the authors.

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
