# Peer review of "Individual and Socioeconomic Contextual Factors Associated with Obesity in Brazilian Adolescents: VigiNUTRI Brasil"

_ijerph, 2022, doi:10.3390/ijerph20010430_

Round 1
Reviewer 1 Report
This study presents useful information, but there are two major points which I propose should be dealt with before it can be published.
A) The models are described as “multilevel” (line 161). But have they actually been fitted using multilevel methods? This is necessary, of course, to take account of the fact that the environmental variables are the same for all adolescents from the same municipality. But there is no information in the Results or in the Statistical Methods section to show that proper multilevel methods have been used.
B) There is insufficient information about the sampling and insufficient discussion of possible biases. On lines 73-74, we learn that the participants in this study were selected because they had “complete data of eating behavior, food consumption and anthropometry”. First issue: we know that 23500 adolescents have these data, but how many do not? Does this selection introduce bias – why does the health care professional complete these data for some but not others? (The discussion of limitations of the study only considers this point in relation to reducing the sample size.) Second issue: this statement about “complete data” is in conflict with the footnote to Table 4, which says that only half the sample (11,928) have the information about consuming three main meals a day. Why? Possible bias?
T: Table 5. As this table contains the main results, I suggest that you should try to make it appear on one pageT
Line 430: I don’t think that “the data is contained in the article” is the correct statement. I believe that you are expected to state here how an interested reader can obtain the actual file of data that you used for the analyses (e.g. by requesting it from you). If it cannot be provided, say why.
Details follow:
Line 12: change “individualized” to “individual” (or alternatively, omit this word altogether)
Line 17: you need to specify what PR stands for
Lines 17-20: write “95% CI” not “CI95%”
Lines 30-31: “the BMI increased more than tenfold” is obviously wrong! Do you mean that the number of adolescents who were obese according to the BMI increased?
Line 68: capital letter for Portuguese
Line 83: “information was obtained” not “it was obtained information”
Line 98: “score” not “scores”
Lines 99-101: delete the word “identified” – just “when” is enough
Line 121: capital letter for “American”
Line 126: change “, stands out the closer to 1” to “. The closer the Index is to 1”
Line 133: delete “Besides”
Line 138: “Covariates” – no apostrophe required
Line 145: “Data cleaning” is a better title than “Bias”. You are talking about errors here, which only lead to bias if they are systematic.
Line 199: write 23,509 not 23.509. You need to make a similar change at many places, including the footnotes of Tables 2, 3 and 4.
Line 203 and elsewhere: why not write “95% CI” in the text, as it is written in the tables? It is easier to read than when 95% is a subscript.
Table 3 (and also 5): “e” appears instead of “and”. But it would be clearer to just write, for example, “0.47 – 0.54” in these situations.
Line 238: capital letter for December
Table 5: Save space by not repeating “degree of urbanization” after “Low”, “Medium” and “High”
Line 272: “individual” not “individualized”
Line 301: “discussed” or “examined” or “studied”, not “pictured”
Line 306: “agree with” or “corroborate”, not “corroborate with”
Lines 331-2: “adolescents who consumed UPF tended to have” not “part of the adolescents who consumed UPF had”
Line 345: “analyses” not “analyzes”
Line 373: “Kelly” not “For Kelly”
Line 374: “point out” not “point”
Line 400: “recorded” not “used”
Line 402: the word “sub-recorded” is not clear. Do you just mean “missing”?
Line 410: “analyses” not “analyzes”
Lines 413-4: write “association of the social …..consumption with the rate” not “association that the social….. consumption exert on the rates”
Lines 415-6: “an efficient tool for periodic use” not “a tool for periodic and parsimonious use”
Reviewer 2 Report
The Authors decided to undertake a crucial problem of factors associated with obesity in adolescents. The topic is widely researched but still very relevant, as excess body weight in adolescence can have an unfavourable impact not only on the current but also on the future well-being of the youth. Additionally, it is a very relevant issue, considering the problem of public health.
The detailed comment and suggestions are provided below:
· Abstract:
“…regarding the anthropometric…” – the word “anthropometric” is an adjective; maybe the Authors meant “anthropometry”?
o “…among those who consumed hamburger…” – judging by the rest of the sentence, it should be “hamburgers”;
o the number of individuals and the age range of the examined population should be provided;
o considering that the abstract should be as straightforward as possible, in the Reviewer’s opinion, the Authors should provide a short explanation of the terms “eating habits” and “food behaviours”, as well as clearly point the direction of the “associations with obesity”;
· Introduction:
o in the reviewer’s opinion, the Authors should provide a few more examples presenting the problem of the prevalence of obesity worldwide (i.e. populations of different countries);
o to the Reviewer’s knowledge, “dyslipidaemia” is generally used only in the singular form;
· Material and methods:
o “…regarding the anthropometric…” – once again, the word “anthropometric” is an adjective; maybe the Authors meant “anthropometry”?
o the Authors should define what is meant by the different levels of density, as in can be interpreted very differently in varying parts of the world;
o the collection of parameters regarding food/ diet is described twice – it is sufficient to provide the description just once;
o in the description of the questions asked in regards to eating habits, the Authors state: “…which are the meals usually eaten by him/her (options are breakfast, morning snack, lunch, afternoon snack, dinner and evening snack)”; does it still apply to the meals eaten in front of the screen (in connection with the previous question) or is it a standalone question designed to know how many meals the participants eat, in general?
o there is no information:
§ on the methods of measuring the body height and weight were measured (tools, method, accuracy, etc.);
§ on the methods of calculating the participants’ age, for example, by determining the exact calendar age etc.;
· Results:
o in l.199, the thousands are divided using period, while previously a comma was used – it should be unified;
o the confidence intervals are provided in two different ways – CI95% or 95%CI, which also should be unified;
· Discussion:
o “In Brazil, there is a great sociocultural, economic and urbanisation inequality among geographic regions…” – in the Reviewer’s opinion, it will be helpful to the readers to provide slightly more extensive information on this topic, as, most likely, not everyone is familiar with this issue;
o “In general, it can be considered that the fact that the traditional Brazilian food pattern (consumption of rice, beans, among other foods) is present in the daily life of the population contributes to the prevention of obesity since it is characterised as a healthier diet” – healthier compared to what?;
o “It is not possible to infer that the consumption of fruits and vegetables, by itself, has a protective effect against the disease” – it is true; however, considering that, it will be helpful to provide short information on how (if not directly) the consumption of fruits and vegetables can protect against excess body mass;
o while listing the study limitations, the Authors should additionally discuss the following:
the limitations of the use of BMI as opposed to body composition measures;
o the fact that the food consumption was assessed only based on one previous day, which can significantly influence the results;
· Conclusions:
o the Authors state that “The persistence of this scenario implies a worsening in the current and future health of adolescents.” – this is true, but the issue of the relationship between excess body mass occurring in childhood/ adolescence and later in life was not discussed in the manuscript; some information on this topic should be provided;
· General remarks:
o the text should be proofread once again, and the English language should be appropriately corrected.
Round 2
Reviewer 1 Report
The authors have answered satisfactorily to the points that I raised in my review of the original submission.